# Automated High-Resolution Time Series Mapping of Mangrove Forests Damaged by Hurricane Irma in Southwest Florida

**Matthew J. McCarthy [1,\*]**, **Brita Jessen [2]**, **Michael J. Barry [3]**, **Marissa Figueroa [2]**, **Jessica McIntosh [2,4]**, **Tylar Murray [1]**, **Jill Schmid [2,4]** and **Frank E. Muller-Karger [1]**

1   Institute for Marine Remote Sensing, College of Marine Science, University of South Florida, 140 7th Ave. South, Saint Petersburg, FL 33701, USA; tylarmurray@mail.usf.edu (T.M.); carib@usf.edu (F.E.M.-K.)
2   Rookery Bay National Estuarine Research Reserve, Florida Department of Environmental Protection, 300 Tower Rd, Naples, FL 34113, USA; brita.jessen@dep.state.fl.us (B.J.); marrisa.b.figueroa@dep.state.fl.us (M.F.); Jessica.mcintosh@dep.state.fl.us (J.M.); jill.schmid@dep.state.fl.us (J.S.)
3   Tatenda, Inc., 5800 SW 188th Ave., Southwest Ranches, FL 33332, USA; mikebarryecologist@gmail.com
4   Center for Coastal Oceans Research, Institute of Water and Environment, Florida International University, Miami, FL 33199, USA
\*   Correspondence: mjm8@mail.usf.edu

**Abstract:** In September of 2017, Hurricane Irma made landfall within the Rookery Bay National Estuarine Research Reserve of southwest Florida (USA) as a category 3 storm with winds in excess of 200 km h$^{-1}$. We mapped the extent of the hurricane's impact on coastal land cover with a seasonal time series of satellite imagery. Very high-resolution (i.e., <5 m pixel) satellite imagery has proven effective to map wetland ecosystems, but challenges in data acquisition and storage, algorithm training, and image processing have prevented large-scale and time-series mapping of these data. We describe our approach to address these issues to evaluate Rookery Bay ecosystem damage and recovery using 91 WorldView-2 satellite images collected between 2010 and 2018 mapped using automated techniques and validated with a field campaign. Land cover was classified seasonally at 2 m resolution (i.e., healthy mangrove, degraded mangrove, upland, soil, and water) with an overall accuracy of 82%. Digital change detection methods show that hurricane-related degradation was 17% of mangrove forest (~5 km$^2$). Approximately 35% (1.7 km$^2$) of this loss recovered one year after Hurricane Irma. The approach completed the mapping approximately 200 times faster than existing methods, illustrating the ease with which regional high-resolution mapping may be accomplished efficiently.

**Keywords:** wetlands; WorldView-2; sunglint; supercomputing; Rookery Bay; National Estuarine Research Reserve (NERR)

## 1. Introduction

Up to 71% of global wetlands have been lost to anthropogenic development (e.g., construction, wetland drainage, hydrologic alterations) during the 20th century, and are expected to continue to decline at a rate of 1%–3% annually [1,2]. A changing global climate (e.g., sea-level rise, altered drought and precipitation patterns, extreme storm events) compounds the stress to these ecosystems. Hurricanes alone cause significant changes to species relative abundance and density through wind damage and storm surge, and may increase in frequency and intensity with a warming Earth [3–5].

Coastal wetlands are estimated to generate over USD$200,000 per hectare per year to local economies and are areas of beneficial nutrient filtering, carbon sequestration, shoreline stabilization, and flood prevention, as well as habitats for numerous species of fish, birds, and invertebrates [6,7]. Further, recent research suggests that better understanding and monitoring of wetland extent is needed to fill vital knowledge gaps in understanding global greenhouse gas emissions [8,9].

Filling these gaps and enhancing management capabilities to manage coastal wetland habitats require accurate and up-to date information on wetland area; habitat composition; biomass and carbon stock; and impacts of stresses such as development, sea level, and temperature changes associated with climate change and severe storms [10,11]. This requires rapid repeat monitoring of coastal habitats at very high spatial resolutions to facilitate local detection of change before it expands beyond the capacity of existing conservation and restoration methods.

Spatial resolution and the quality of satellite-based remote sensing for land cover assessments have increased substantially in recent years with technological advances that result in better precision and accuracy [10–14]. Today, most repeated resource maps for wetlands covering large areas are derived using Landsat/Sentinel-2 satellite sensors (i.e., order of 10–30 m pixels) [15,16]. Much higher spatial resolution data are increasingly available from commercial sources. There is substantial burden associated with these data in addition to their high cost, including the data processing requirements (i.e., data transfer time, storage capacity, and computation time), thereby necessitating the development of techniques that leverage the computational efficiency of supercomputers.

Automated mapping methods to process large numbers of commercial satellite data to cover large areas or time series have not been implemented primarily owing to several constraints. These include the cost of the imagery; access to, transfer, and storage of large volumes of imagery; lack of sufficient training data for automated thematic classifications; and inefficient processing methods. Two-meter resolution Level 1B WorldView-2 imagery, for example, costs approximately USD$4000 and take up ~300 MB per scene covering ~260 km$^2$. When processed to Level 2A (i.e., atmospherically corrected and converted to surface reflectance) as floating point rasters, file size can increase by an order of magnitude. Good data to train thematic classification algorithms are typically collected in the field (e.g., ground-truth points), or by labeling target map classes on pre-existing multispectral imagery. Both are time-consuming and may be prohibitively expensive for large-scale or time-series mapping of diverse habitats. For example, a recent effort to map some 65,000 km$^2$ of coastal shallow-water habitats using remote sensing required 10 years of field surveys [17]. The processing of various thematic classification algorithms, mapping the images to specific geographic projections, and stitching different images together into a larger map can take hours to days and require intensive manual work [18].

In this study, we outline a strategy to streamline this process and generate mapped wetland land cover classes based on several dozen cases of high-resolution commercial imagery frequently, accurately, and efficiently. We describe the results of a project to map and monitor mangrove forest change from 2010 to 2018 using 91 high-resolution WorldView-2 satellite images. We evaluate changes as a result of chronic stress owing to sea-level rise or hydrologic alteration, as well as those resulting from Hurricane Irma, a category 3 storm that made landfall within the study area in September of 2017. The goals of this study were to (1) generate a time series of high resolution land cover maps through automated mapping techniques; (2) distinguish between healthy- and degraded-mangrove and non-mangrove vegetation; and (3) evaluate mangrove decline, recovery, and die-off location and extent in the context of coastal habitat management with the assistance of local-knowledge end users, including environmental managers. The motivation for this project was to address the explicit management need of identifying the location, extent, and drivers of mangrove loss with the Rookery Bay study area.

The study area is located on the southwest coast of Florida (USA) along the Gulf of Mexico, and includes three management jurisdictions (Rookery Bay National Estuarine Research Reserve (NERR), Ten Thousand Islands National Wildlife Refuge, and Collier-Seminole State Park; Figure 1). The area is actively managed and includes the location where Hurricane Irma made landfall as a

category 3 hurricane with winds in excess of 200 kph after passing through the Florida Keys in September of 2017. Located in a subtropical climate zone, the reserve spans over 445 km$^2$ of low-lying mangrove-dominated coastal plain adjacent to the high-density residential development of Marco Island. Rookery Bay NERR is replete with mangrove rookery islands that were damaged by the storm (Figure 2), as well as marsh, upland, and salt flats, and is adjacent to the Everglades National Park.

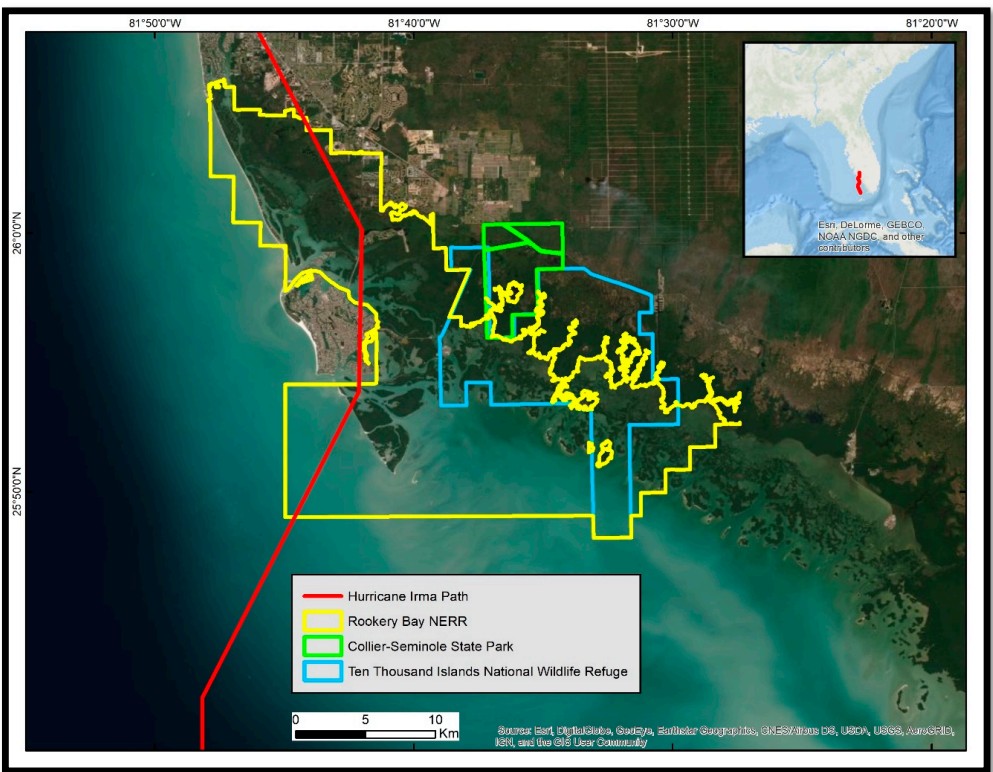

**Figure 1.** Southwest Florida, USA, showing the path of Hurricane Irma (red line in main graphic and in the inset showing the western Gulf of Mexico and the state of Florida; ArcGIS Basemap Source: ESRI). Local management jurisdiction boundaries are shown in different colors. NERR, National Estuarine Research Reserve.

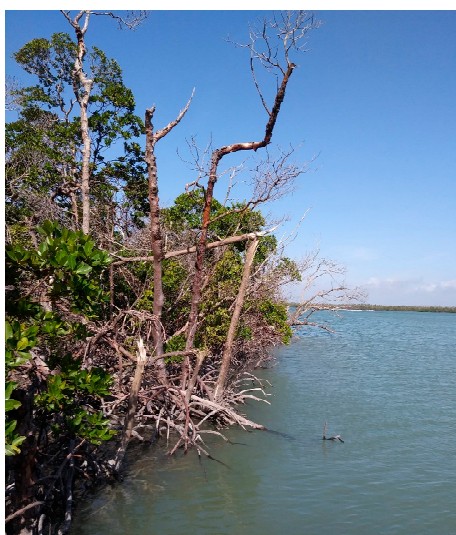

**Figure 2.** Photograph of broken and defoliated red mangroves from the January 2019 site visit.

## 2. Materials and Methods

This project used 91 WorldView-2 satellite images acquired in 2010 (3), 2016 (2), 2017 (36), and 2018 (50) to map a time-series of the study area. The WorldView-2 satellite sensor has been operated by DigitalGlobe[TM] since it was launched in 2009 to collect data in eight multispectral bands (Table 1). Up to 1 million km$^2$ of land area is imaged daily at a nominal resolution of two meters per pixel (DigitalGlobe[TM], 2009). Land cover classifications completed using WorldView-2 imagery demonstrated superior accuracy over similar high-resolution satellite sensors with fewer spectral bands [14,19,20]. Images were obtained under a federal research grant licensing agreement with support of the U.S. National Science Foundation and through a partnership with the Polar Geospatial Center (PGC, University of Minnesota). Images were obtained in the Level-1B National Imagery Transmission Format (NITF) with metadata.

**Table 1.** WorldView-2 imagery specifications (NIR stands for near-infrared; Digital Globe™, 2009).

| Band Name | Band Number | Center Wavelength (nm) | Band Coverage (nm) | Effective Bandwidth (nm) |
|---|---|---|---|---|
| Coastal | B1 | 427 | 396–458 | 47.3 |
| Blue | B2 | 478 | 442–515 | 54.3 |
| Green | B3 | 546 | 506–586 | 63.0 |
| Yellow | B4 | 608 | 584–632 | 37.4 |
| Red | B5 | 659 | 624–694 | 57.4 |
| Red Edge | B6 | 724 | 699–749 | 39.3 |
| NIR I | B7 | 833 | 765–901 | 98.9 |
| NIR II | B8 | 949 | 856–1043 | 99.6 |

Data were stored on a single Redundant Array of Independent Disks (RAID-enabled) disk server accessed by a supercomputer with over 4000 processing nodes. Disk input/output transfer times were not included in processing time estimates as this is variable depending on network and storage access configuration.

The preparation of the images for thematic classification was completed using Matlab[TM] software. This included the following: radiometric calibration, Rayleigh atmospheric scattering correction, remote sensing reflectance (Rrs) computation, classification, and smoothing (see McCarthy [21] for detailed description).

Additionally, some scenes contained specular reflection ("sunglint") off water surfaces. These preclude the accurate identification and classification of many pixels as "water". Removing sunglint from satellite images is common practice [22,23], but has not been implemented in high-resolution images in an automated fashion to our knowledge. To deglint our images, we identified the spectral signature of glinted pixels and applied a regression-based correction using all water pixels in the image. Rrs images were evaluated for spectral signatures of glint and glint-free water pixels. We identified a sun glint spectral signature (Figure 3) across scenes, whereby reflectance values in bands from one array were consistently lower or higher than adjacent bands from the other array. This pattern was robust for bands 3–8, but not for the coastal or blue bands (i.e., bands 1 and 2).

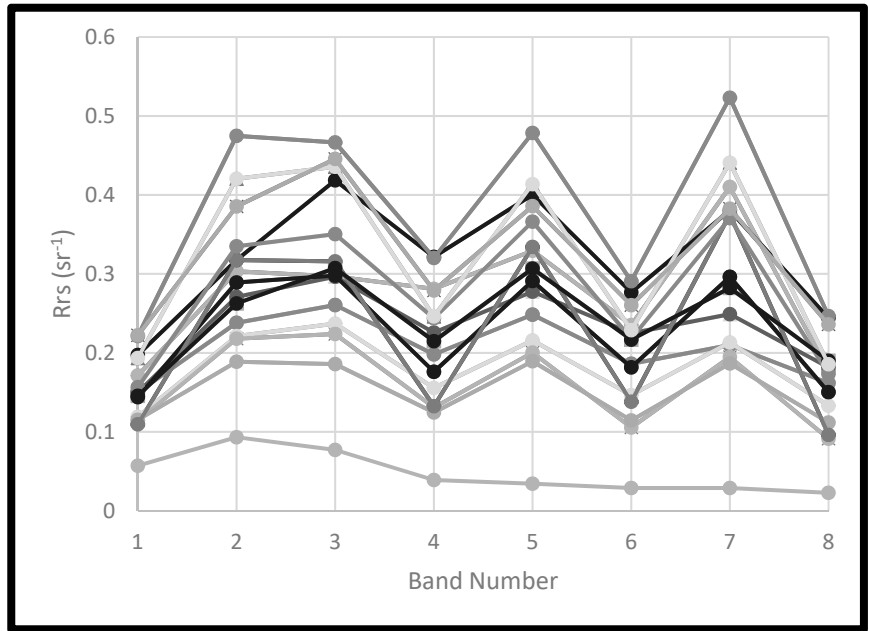

**Figure 3.** Spectral signatures of a sample of glinted pixels (upper, multiple), and the spectral signature of a glint-free pixel from the same scene (lower, single).

The algorithm implemented to detect glint consisted of a loop that identified all glinted-water pixels and distinguished them from pixels that exhibited a typical water signature. This approach thus uses all water pixels from the image, as opposed to a sample. The dual-array nature of the WorldView-2 sensor (i.e., one array captures light for bands 1, 4, 6, and 8; the other captures light in bands 2, 3, 5, and 7) requires that the six WorldView-2 visible-wavelength multispectral bands (i.e., bands 1–6) be corrected relative to their respective near-infrared (NIR) band. The ambient NIR value for each array was selected as the minimum from NIR I (B7) and NIR II (B8) of the glint-free water pixels. Linear regressions were performed between each multispectral band and its respective NIR band on all water pixels. A new loop was performed to deglint all water pixels. Pixels were deglinted using the following equation from [22]:

$$Rrs'_i = Rrs_i - b_i(Rrs_{NIRj} - Min_{NIRj}), \tag{1}$$

where $Rrs'_i$ is the deglinted pixel value in band i, $Rrs_i$ is the reflectance value in band i, $b_i$ is the regression slope, $Rrs_{NIRj}$ is the target pixel's NIR value for the appropriate sensor array (j), and $Min_{NIRj}$ is the respective minimum NIR value.

We used a decision tree classifier within the automated processing protocol to arrive at a predefined set of thematic land cover classes. The decision tree is a multi-stage classifier tool that uses a series of binary decisions to assign a thematic class to each image pixel. Decision tree classifiers built to use spectral libraries and historical data are routinely used for global-scale mapping of coarse imagery [16].

This decision tree was built to map five classes: healthy mangrove, degraded mangrove, upland (non-mangrove), developed and bare (e.g., buildings and soil), and water. The classification was modified from that defined in [21] to include degraded mangrove. The degraded-mangrove node was developed based on interpretation of spectral signatures collected from WorldView pixels of areas that were ground-validated during a January 2018 field visit. Degraded mangroves were characterized as displaying partial to complete defoliation relative to healthy mangroves, and could include broken branches or trunks. The criteria were that the pixels contain vegetation, identified with the normalized difference vegetation index, and a value of 0–0.12 in the NIR I band.

We ran both the Python and Matlab functions on a supercomputer cluster at the University of South Florida (USF). This facility has over 4000 processors with access to a combined 2.5 terabytes of

memory. The codes used were developed on a standard desktop or laptop computer, but the cluster allowed us to process all 91 images in parallel. This protocol can be run on a standard computing environment, but will be limited in batch-processing efficiency by the number of images that may be run simultaneously.

Ground reference points (GRPs; 3 m horizontal accuracy) were collected throughout the reserve during surveys conducted from September to December of 2018. Transects were walked as feasible, and data were collected as polylines for homogenous habitats (i.e., a new polyline was started when one habitat transitioned to a different habitat). Photographs and detailed comments were included in the data collection protocol. Polylines were then converted to points in ArcMap 10.1 at 20 m intervals to avoid spatial autocorrelation [20]. A total of 2700 GRPs were collected. Quality control of GRPs required the following criteria: located within the extent of the fall 2018 WorldView-2 mosaic; located at least 20 m from each other to avoid spatial autocorrelation; and clearly representative of healthy mangrove, degraded mangrove, upland, bare soil, or water thematic classes. Mangroves were defined as belonging to the red (*Rhizophora mangle*), white (*Laguncularia racemosa*), or black (*Avicennia germinans*) species. Given these criteria, a total of 992 GRPs were selected as valid. These included 50 "water" points that were manually digitized via aerial reference imagery (Figure 4).

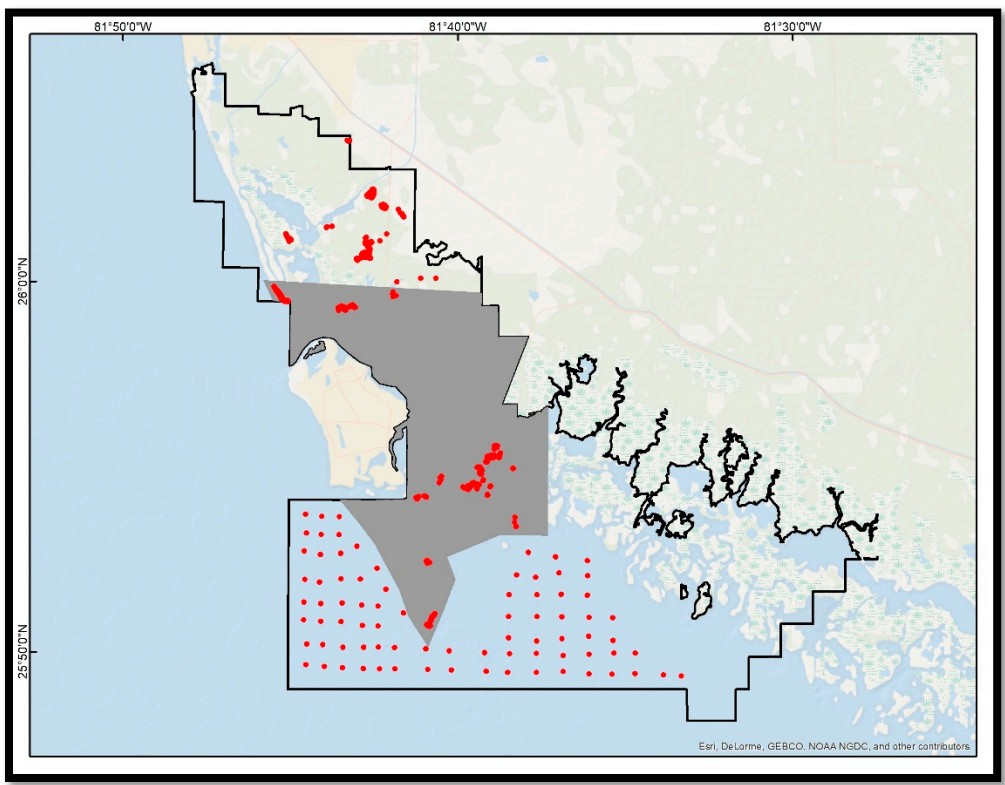

**Figure 4.** Ground-reference points collected within the Rookery Bay NERR (jurisdiction boundary shown in black) to validate the classified maps (ArcGIS Basemap Source: ESRI). Area of overlap between annual maps that was used for change detection is displayed in grey.

All classified images were mosaicked by year (i.e., 2010, 2016, 2017, and 2018) into seamless maps in ArcMap 10.1 using the "Mosaic to New Raster" tool. Additionally, images were mosaicked seasonally for the following periods during which sufficient imagery was available to cover a substantial portion of the reserve: spring 2016; spring 2017; winter 2017; and fall 2018. The mosaicked maps were then clipped to match the extent of the reserve.

Map accuracy was calculated for the fall 2018 map as it corresponded to the field campaign period. This is the only period for which ground-truth data were available, and so the results are assumed to be

representative of all maps. Single-date validation is not the most robust means for assessing accuracy when change detection is involved, but is often the only available approach when assessing natural disaster impacts [24,25]. Using field surveys, we attempt to compensate for the single-date validation issue by acquiring the most accurate validation data [26]. Accuracy was calculated in ArcMap by intersecting the GRPs with the classified-map pixels and recording the number of pixels that agreed and disagreed with validation data. User's and producer's accuracies were calculated for each class, as well as overall accuracy [27].

## 3. Results

### 3.1. Mapping Efficiency

All 91 WorldView-2 images were mapped on the USF supercomputer. Computation time was approximately 6 minutes per image (9 h total for 91 images). The authors previously conducted this work using complimentary tools and methods in ENVI 5.4 software for a processing time of approximately 20 h per image, corresponding to approximately 76 days for 91 images.

### 3.2. Mapping Accuracy Assessment

Map accuracies for the fall 2018 mosaic are shown in Table 2. The results include overall map accuracy, producer's accuracies, and user's accuracies.

**Table 2.** Confusion matrix for decision tree (DT) classifier results for the fall 2018 map mosaic indicating the number of ground reference points (GRPs) classified correctly or incorrectly. Overall accuracy in bold (Kappa = 0.757).

|  |  | Reference |  |  |  |  |  |  |
|---|---|---|---|---|---|---|---|---|
|  |  | Soil | Degraded Mangrove | Healthy Mangrove | Upland | Water | Total | User's Accuracy |
|  | Soil | 316 | 4 | 0 | 4 | 10 | 334 | 95% |
|  | Degraded Mangrove | 4 | 28 | 14 | 0 | 4 | 50 | 56% |
|  | Healthy Mangrove | 3 | 17 | 256 | 52 | 0 | 328 | 78% |
| Classified | Upland | 0 | 3 | 61 | 133 | 0 | 197 | 68% |
|  | Water | 0 | 0 | 0 | 0 | 83 | 83 | 100% |
|  | Total | 323 | 52 | 331 | 189 | 97 | 992 |  |
|  | Producer's Accuracy | 98% | 54% | 77% | 70% | 86% |  | **82%** |

### 3.3. Change Detection

The surface area occupied by each land cover class was calculated for each time period. Change in area between periods was estimated by difference and indicates that soil and degraded mangrove classes increased in area by a combined 8.6 km$^2$, while upland, healthy mangrove, and water decreased by the same amount (Table 3). Unfortunately, there was limited imagery available for some time periods. Therefore, change in area was only evaluated for a portion of the reserve that was consistently covered for the following seasonal maps: spring 2016; spring 2017; winter 2017; and fall 2018. This portion of the reserve represented only 14% of the total reserve area, but it helps understand change due to a major storm like Hurricane Irma.

**Table 3.** Class coverage area (km$^2$).

|  | Spring 2016 | Spring 2017 | Winter 2017 | Fall 2018 | 2016–2018 Change |
|---|---|---|---|---|---|
| Soil | 5.00 | 4.16 | 3.17 | 8.85 | 3.84 |
| Degraded Mangrove | 0.44 | 2.38 | 9.92 | 5.21 | 4.77 |
| Upland | 4.06 | 2.64 | 1.22 | 2.30 | −1.77 |
| Healthy Mangrove | 29.94 | 26.33 | 21.36 | 23.09 | −6.85 |
| Water | 25.27 | 29.13 | 29.04 | 25.24 | −0.03 |

## 4. Discussion

Mangrove forests along southwest Florida have been migrating inland or dying off owing to a combination of chronic sea-level rise and acute damage by severe storms, in particular by Hurricane Irma [5,28,29]. Managers within the Rookery Bay NERR have engaged in mangrove restoration and conservation efforts guided by a baseline map from 2010, but need updated maps to identify areas vulnerable to or impacted by chronic and acute stressors [28,30]. The goal of this research was to assist in this process by generating an updated series of maps to evaluate degraded and healthy mangrove stands relative to other habitats, and quantify change in these habitats over the period spanning 2010–2018.

By developing automated mapping methods run on a supercomputer, we demonstrated the capacity of advanced computational technology to complete habitat mapping approximately 200 times faster than traditional methods (i.e., 6 minutes per image vs. 20 h per image). Using Python and Matlab programming languages, we processed 91 high-resolution WorldView-2 satellite images, including projection, radiometric calibration, atmospheric correction, Rrs conversion, mapping, and smoothing, in approximately 9 h to produce a total of six maps representing a nine-year time-series of coastal habitat coverage.

A three-month field campaign was conducted to coincide with and validate the fall 2018 map. The 992 ground-truth points collected indicated that the overall map accuracy was 82%. Degraded mangrove was the least-accurate class (58%), likely because this class represents mangrove stands that were partially defoliated, broken, and otherwise damaged by wind and storm surge, yet maintained a weak vegetated spectral profile. It included sub-canopy signatures of sediment, marl overwash, or hurricane debris. The confusion matrix (Table 2) indicates that most of the misclassification of degraded mangrove was the result of confusion with healthy mangrove, which is likely a result of the NIR threshold applied to distinguish between the two. Using a threshold creates a binary classification that does not necessarily capture the "degradation gradient" between the two classes. Future research should consider applying spectral unmixing to better distinguish between these classes. Patterns of degraded mangrove (Figure 5) relative to other vegetation appear consistent with visual, on the ground observations of damage caused by the storm.

Because of the lack of consistent coverage between the different seasonal periods, we focused our pre- and post-Hurricane Irma change detection analysis on a 65 km$^2$ area near the center of the reserve (Figure 4). We observed a net loss of 6.85 km$^2$ (22%) of mangrove forest from spring 2016 to fall 2018. Of this, 4.97 km$^2$ was lost in the six-month period after Hurricane Irma. In the following 6 months, a total of 1.73 km$^2$ of mangroves recovered, but 1.6 km$^2$ was converted to bare soil. We interpret the latter as mangrove die-off, because it indicates that mangrove stands remained defoliated for at least one full growth season following the initial hurricane-induced defoliation that occurred after the September 2017 hurricane. These patterns are consistent with those observed by [5], which described both short-term and delayed mortality of mangroves as a result of wind-induced defoliation, and marl-overwash-deposit root smothering, respectively, based on field surveys of mangrove forests within and adjacent to Rookery Bay NERR. Indications that mangrove forests recovered during the growing season were confirmed by findings of [5], that a 40%–70% reduction in mangrove-canopy cover by wind-induced defoliation facilitated the recruitment of mangrove saplings. Upland vegetation results follow a similar pattern that we interpret as hurricane-induced loss followed by some recovery and some die-off. Seasonal phenology is unlikely to be a substantial driver in these results as the region experiences a subtropical climate with negligible winter senescence of wetland or upland vegetation.

The synoptic, seasonal maps of recovery and die-off patterns (Figure 6) will be used by reserve managers to prioritize mangrove conservation and restoration efforts. Efforts to map global tropical wetlands have relied on medium-resolution or coarser satellite imagery, and required years of processing to complete [15]. Further, detection of change in 10–30 m pixels covering 100–900 m$^2$ each may be insufficient for identifying gain or loss before irreversible changes have already occurred. Addressing these challenges will require accurate, global, automated mapping of high-resolution images.

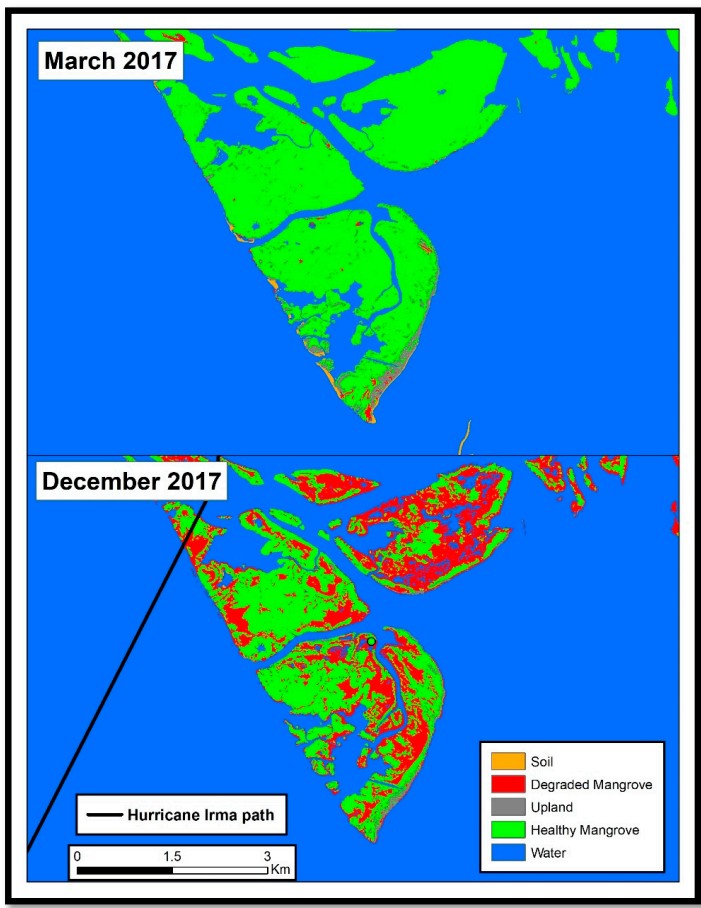

**Figure 5.** Damage to mangroves caused by Hurricane Irma in September 2017.

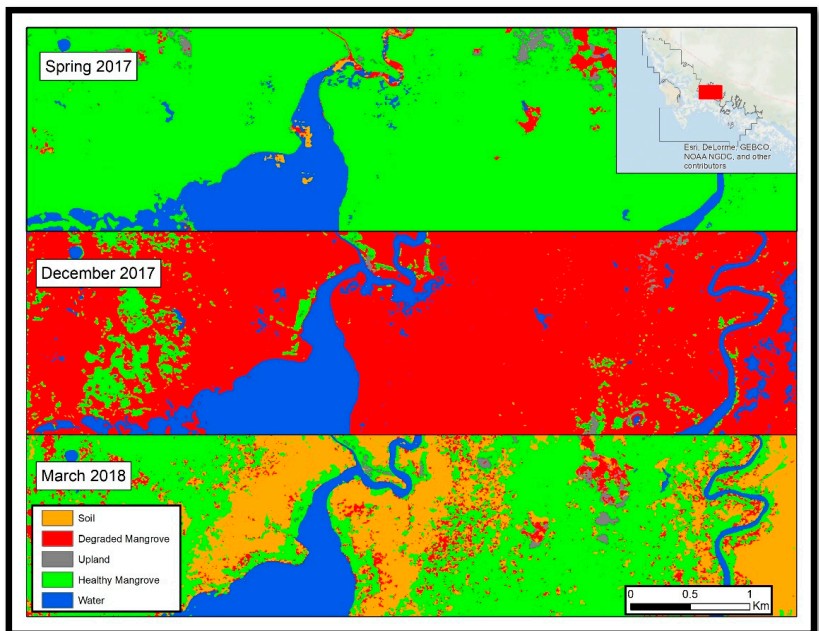

**Figure 6.** Subset of the study area (inset map) reflecting various stages of mangrove change. Healthy mangrove dominated the reserve in early 2017 (top), but was largely degraded following Hurricane Irma (center), and either recovered or died by the following spring (bottom).

## 5. Conclusions

Automated processing and mapping of 91 2 m resolution satellite images facilitated the evaluation of damage caused by Hurricane Irma in 2017 to a mangrove forest in southwest Florida. Approximately one-fifth of the mangrove forest in the study area was damaged. Of that, one-third regenerated to healthy mangrove, one-third died off, and the remainder converted to other classes.

As satellite and computational technologies advance, mapping at higher resolutions over larger scales and monitoring with time series will become much needed standard practice for conservation and resource management, among many other applications. We demonstrated the feasibility of mapping of nine years of seasonal and annual imagery approximately 200 times faster than existing single-image methods. Within this approach, we included a novel automated method for deglinting imagery that would otherwise inaccurately identify water, and that may contribute to further research in water quality, benthic mapping, and others. Efforts to carry out accurate and efficient mapping must consider the tradeoffs and feasibility of the available method and advance them with the considerations noted here.

**Author Contributions:** Conceptualization, M.J.M., B.J., and F.E.M.-K.; methodology, M.J.M., B.J., and M.J.B.; software, T.M. and M.J.B.; validation, M.J.M.; formal analysis, M.J.M., B.J., M.F., J.M., and J.S.; resources, B.J., M.J.B., J.M., and F.E.M.-K.; data curation, M.J.M. and T.M.; writing—original draft preparation, M.J.M.; writing—review and editing, B.J., M.J.B., M.F., J.M., T.M., J.S., and F.E.M.-K.; visualization, M.J.M. and J.S.; supervision, B.J. and F.E.M.-K.; project administration, M.J.M. and F.E.M.-K.; funding acquisition, M.J.M. and F.E.M.-K. All authors have read and agreed to the published version of the manuscript.

**Funding:** This work was sponsored by the National Estuarine Research Reserve System Science Collaborative, which supports collaborative research that addresses coastal management problems important to reserves. The Science Collaborative is funded by the National Oceanic and Atmospheric Administration and managed by the University of Michigan Water Center (NAI4NOS4190145). This work was also funded by NSF grant number 1728913. Geospatial support for this work provided by the Polar Geospatial Center under NSF PLR awards 1043681 and 15559691. Partial support was provided also by NASA grant NNX14AP62A 'National Marine Sanctuaries as Sentinel Sites for a Demonstration Marine Biodiversity Observation Network (MBON)' and NOAA/ONR grant NA19NOS0120199 'Implementing a Marine Biodiversity Observation Network (MBON) in South Florida to Advance Ecosystem-Based Management', both funded under the National Ocean Partnership Program (NOPP RFP NOAA-NOS-IOOS-2014-2003803 in partnership between NOAA, BOEM, and NASA), and the US Integrated Ocean Observing System (IOOS) Program Office. This manuscript is a contribution to the Marine Biodiversity Observation Network (MBON).

**Conflicts of Interest:** The authors declare no conflict of interest. The funders had no role in the design of the study; in the collection, analyses, or interpretation of data; in the writing of the manuscript; or in the decision to publish the results.

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
