# Peer review of "Automated High-Resolution Time Series Mapping of Mangrove Forests Damaged by Hurricane Irma in Southwest Florida"

_remotesensing, doi:10.3390/rs12111740_

Round 1

Reviewer 1 Report

Dear authors,

The largest issue is that no accuracy assessment for the pre-2018 vegetation classifications has been done, which puts the change detection results themself on very weak legs. Congalton & Green wrote a very good paper about the importance of accuracy assessment for change detection around 1999. I recommend you have a look at it.

Further, maybe add a map for the footprints of the annual coverage of the WV-2 images to show where they overlap.

Please find my further specific comments in the attached PDF.

Looking forward to the next version of the paper.

Author Response

Reviewer 1

Dear authors,

The largest issue is that no accuracy assessment for the pre-2018 vegetation classifications has been done, which puts the change detection results themself on very weak legs. Congalton & Green wrote a very good paper about the importance of accuracy assessment for change detection around 1999. I recommend you have a look at it.

                The reviewer is correct. Professor Congalton wrote numerous articles and books detailing the importance of robust accuracy assessments, especially in studies of change. He also writes in those publications that the accuracy of validation data is of utmost importance. For this study, we attempted to gather the most accurate validation data possible (i.e. through field survey as opposed to image interpretation) – an approach that effectively required us to sacrifice multi-temporal validation-data acquisition in favor of highly accurate single-date validation. While we agree that this approach is not ideal, we feel that it still offers value, and we have added lines 220-223 to acknowledge these potential issues. We also include the noted Macleod/Congalton reference, as well as two recent papers that attempt to map hurricane damage and assess accuracy that state that when the change under scrutiny is caused by natural disasters it is generally difficult to acquire either in-situ reference data for before-and-after time periods, or cloud-free independent reference imagery, which requires the analyst to rely on single-date validation. We hope that these justifications are sufficient enough to warrant the use of this high-validity single-date assessment.

Further, maybe add a map for the footprints of the annual coverage of the WV-2 images to show where they overlap.

                We agree, although in lieu of adding multiple figures indicating the total coverage, we added the overlapping outline to Figure 4.

Please find my further specific comments in the attached PDF.

                We have addressed these comments with responses in the PDF and made appropriate changes to the text as seen in Tracked Changes.

Reviewer 2 Report

The authors did a good job at describing an innovative and automatic way to utilize WorldView-2 satellite images to assess the damage induced by Hurricane Irma. The overall accuracy of this method is very promising and turns out significantly faster than traditional methods. I don't have any major concerns with this study. Here are some minor comments for the authors:

  1. Looks like the efficiency of this method highly relies on HPC. Would there by any way to scale this in a normal computing environment?
  2. The use of this method is specifically limited to detecting the damaged forests. Can you discuss its potential applications in actual flood detection?

Author Response

Reviewer 2

The authors did a good job at describing an innovative and automatic way to utilize WorldView-2 satellite images to assess the damage induced by Hurricane Irma. The overall accuracy of this method is very promising and turns out significantly faster than traditional methods. I don't have any major concerns with this study. Here are some minor comments for the authors:

Looks like the efficiency of this method highly relies on HPC. Would there by any way to scale this in a normal computing environment?

Yes, this protocol can be run on any standard computing environment. The efficiency is indeed modulated by the number of images that can be processed simultaneously, which is in turn limited by the capacity of the computing environment (e.g. number of cores). The HPC used simply offers many cores. We added lines 192-194 to address this observation.

The use of this method is specifically limited to detecting the damaged forests. Can you discuss its potential applications in actual flood detection?

The reviewer makes a good point given the high accuracy of the results in the water class. However, we did not assess the accuracy of ephemeral water bodies (i.e. temporary floods), the spectral profiles of which could be substantially different from the profiles we used to identify water depending on depth, bottom substrate, and water column properties, so we do not feel that we could justifiably comment on that potential application.

Reviewer 3 Report

First of all, I want to congratulate the authors. The methodological approach that the authors bring to us is a clear improvement for evaluation of  ecosystem damage and recovery using automated techniques and validated with a field campaign.

This paper is well documented and well written with an interesting example from the southwest coast of Florida (USA) along the Gulf of Mexico.

I found just a few minor issues with this paper:

  1. At the end of the first section (Introduction) more information should be added regarding Hurricanes Irma and the studied area, in this case Southwest coast of Florida (USA) along the Gulf of Mexico. If there are, images with the affected mangrove area can be inserted.
  2. At Figures 1 and 3 are missing references.
  3. In table 3, I consider necessary to explain the positive and negative values found in the last column.
  4. Reference no. 24 is missing; I think at line 223 it was included.

“223 vulnerable to or impacted by chronic and acute stressors [23,25]. The goal of this research was to”. Maybe it is “[23-25]”.

  1. The conclusion is very brief. Considering the numerous results that you have obtained, I consider that it is necessary that this section of conclusions be more detailed based on the results obtained from the Results and Discussion Sections.

I think this work is a very good and interesting approach offers the possibility to improve the automated mapping techniques.

Author Response

Reviewer 3

At the end of the first section (Introduction) more information should be added regarding Hurricanes Irma and the studied area, in this case Southwest coast of Florida (USA) along the Gulf of Mexico. If there are, images with the affected mangrove area can be inserted.

We agree and have added lines 91-99 further describing the subject matter, and Figure 2 from the field of damaged mangroves.

At Figures 1 and 3 are missing references.

We agree and thank the reviewer for catching this oversight. We have added appropriate references to both figures citing the ESRI basemap data used.

In table 3, I consider necessary to explain the positive and negative values found in the last column.

We agree and added lines 241-243 to clarify these results.

Reference no. 24 is missing; I think at line 223 it was included.

“223 vulnerable to or impacted by chronic and acute stressors [23,25]. The goal of this research was to”. Maybe it is “[23-25]”.

This reference can be found in line 225: “… in particular by Hurricane Irma [5,23,24].”

The conclusion is very brief. Considering the numerous results that you have obtained, I consider that it is necessary that this section of conclusions be more detailed based on the results obtained from the Results and Discussion Sections.

While we prefer to keep the Conclusions section relatively brief, we agree that we should have included a more wholistic summary and have added relevant conclusions regarding the mapping efficiency, and deglinting algorithm findings (lines 312-316).

Round 2

Reviewer 1 Report

Dear authors,

thank you for considering the comments on the earlier m/s versions. I only have some minor comments on this current version. These are provided in the attached commented PDF version.

Author Response

We have addressed each of the reviewer's comments within the attached PDF document. We thank the reviewer for their thoughtful and thorough consideration of our work.

This manuscript is a resubmission of an earlier submission. The following is a list of the peer review reports and author responses from that submission.